# Cytoplasmic Male Sterility Incidence in Potato Breeding Populations with Late Blight Resistance and Identification of Breeding Lines with a Potential Fertility Restorer Mechanism

**DOI:** 10.3390/plants11223093

**Published:** 2022-11-14

**Authors:** Monica Santayana, Mariela Aponte, Moctar Kante, Raúl Eyzaguirre, Manuel Gastelo, Hannele Lindqvist-Kreuze

**Affiliations:** International Potato Center (CIP), Lima 15024, Peru

**Keywords:** hybrid potato, fertility restoration, acetocarmine, cytoplasmic factors, in vivo pollen tube germination

## Abstract

Cytoplasmic male sterility (CMS) in potato is a common reproductive issue in late blight breeding programs since resistant sources usually have a wild cytoplasmic background (W or D). Nevertheless, in each breeding cycle male fertile lines have been observed within D- and T-type cytoplasms, indicating the presence of a fertility restorer (*Rf*) mechanism. Identifying sources of *Rf* and complete male sterility to implement a CMS–*Rf* system in potato is important since hybrid breeding is a feasible breeding strategy for potato. The objective of this study was to identify male fertile breeding lines and potential *Rf* candidate lines in the CIP late blight breeding pipeline. We characterized male fertility/sterility-related traits on 142 breeding lines of known cytoplasmic type. We found that pollen viability is not a reliable estimate of male sterility in diverse backgrounds. Breeding lines of the T-type cytoplasmic group had higher levels of male fertility than breeding lines of the D-type cytoplasmic group. With the help of pedigree records, reproductive traits evaluations and test crosses with female clones of diverse background, we identified four male parental lines segregating for *Rf* and three female parental lines that generated 100% male sterile progeny. These identified lines and generated test cross progenies will be valuable to develop validation populations for mitochondrial or nuclear markers for the CMS trait and for dihaploid generation of *Rf*+ lines that can be later employed in diploid hybrid breeding.

## 1. Introduction

Potato diploid hybrid breeding requires a harmonic interaction of breeder tools as haploid induction-, *Sli* based-self compatibility, true potato seed management and a cytoplasmic male sterility/fertility restorer system (CMS–*Rf*) [1,2,3,4,5,6,7,8,9]. Developing a CMS–*Rf* in potato is paramount because dihaploids are mostly male sterile even after *Sli* introgression, and this hampers self-compatibility and the obtention of homozygous lines.

Cytoplasmic male sterility (CMS) is an important agronomic trait resulting from the interaction of the organellar and nuclear genomes affecting the production of viable pollen or the correct development of important reproductive structures. CMS is a valuable trait for hybrid breeding because it allows the production of hybrid seed without emasculation from the female counterpart [10,11,12]. Potato CMS have been associated with three types of cytoplasm. T-type cytoplasm male sterility shows low pollen viability, shriveled or indehiscence anthers and is related with either the presence of a fertility restorer gene (*Rf*) or the recessive nuclear alleles for male sterility, ms/ms [13,14]. W/γ-type cytoplasm, associated with *S. stoloniferum*, causes the strongest sterilizing effect characterized by tetrad or lobed microspores; this type of sterility was renamed to T–CMS, tetrad cytoplasmic male sterility [15]. D-type cytoplasm produces stainable but non-functional pollen grains [10,14,16,17,18]. The presence of male fertile genotypes within D-type cytoplasm is assumed by the presence of *R1* and *R3a*, *S. demissum* race specific genes in T-type or W/γ-type genotypes [17].

The capacity of *Rf* genes to encode mitochondria-targeted pentatricopeptide repeats (*PPR*) proteins was first discovered in petunia [19]. The *PPR* gene family encodes proteins that act on mitochondrial transcripts and directly affect the CMS phenotype [20]. Anisimova et al. [21] proposed an *Rf* candidate gene for potato after detecting polymorphisms in the repeated motifs of a fertility restorer like *PPR* (*RFL*–*PPR*) genes associated with sterile and fertile phenotypes in potato of D and T cytoplasmic type. Sterile/fertile phenotypes were successfully differentiated in W/γ type by developing the cleaved amplified polymorphic sequence (CAPS) marker Int2NAD2/Sse9I for an intron of the mitochondrial *nad2* gene [17] or an 859-bp mtDNA fragment between the genes for the ribosomal proteins rpl5 and rps10 [22].

Potato has six cytoplasmic types (T, D, P, A, M and W) based on the combination of its plastid and mitochondrial DNA [18,23,24]. The T type is predominant in the common potato *S. tuberosum*; the D type is related to derivatives of *S. demissum*; the P type derives from phureja group; the A type is typical of *S. tuberosum* Andigenum; the M cytoplasmic type is related to ancestral Andean potatoes and the W type is related to wild species [18].

Mihovilovich et al. [14] observed a genetic bottleneck by screening 978 potato genotypes from CIP main breeding populations. There were increasing proportions of D-type cytoplasm within the late blight resistance population B3. The proportion of B3 breeding lines carrying the D-type cytoplasm increased from 25% to 81.5% as compared to the initial population (introduced varieties or breeding lines and Peruvian varieties) and initial selections (C0). Although few identified male fertile genotypes allowed new recombination at each breeding cycle, a systematic reproductive characterization of breeding lines would favor higher rates of successful crosses, thus a better allocation of resources. In this study we phenotyped the reproductive traits involved in male fertility/sterility of the current cycle of potato late blight resistance breeding lines and validated them by test crosses to identify potential male fertile cytoplasmic lines or bearers of the *Rf* nuclear gene.

## 2. Results

### 2.1. Cytoplasmic Type Composition of Late Blight Resistant Populations

All 142 breeding lines used in this study could be traced back to 13 maternal ancestors. Nine lineages belonged to the D-type cytoplasm group, three to the T-type group and one to the W/γ-type cytoplasm group (Table 1). The B3C3 population was composed of 96.8% clones with the D-type cytoplasm and 3.2% of clones with T-type cytoplasm. The LBHT population was composed of 39.5, 58.1 and 2.3% of clones with the D-, T-and W/γ-type cytoplasm, respectively. The LBHT x LTVR population had 16.2% D-type cytoplasm and 83.8% T-type cytoplasm (Table 1).

### 2.2. Effects of Cytoplasmic Type on Reproductive Traits

All breeding lines produced flowers and had the peak in flowering approximately at 90 days after transplanting. Most tested clones flowered profusely (Figure 1), and no differences were found in the flowering degree for the different cytoplasmic types (*p* = 0, 0.2378, Appendix A). However, breeding lines with D-type cytoplasm produced significantly more pollen than T-type cytoplasm (Figure 1, Appendix A). The only breeding line from W-type cytoplasm produced less than 50 μL of pollen (Appendix A).

The most common floral abnormality in the breeding lines was anther-lobe overlap (*lo*, Figure 1d and Figure 2a,e), as observed in 48 and 75% of the lines with D- and T-type cytoplasms, respectively, and in the only W/γ-type line (Figure 2e). Additionally, D-type breeding lines also showed anther-corolla fusion (ACF, Figure 1d and Figure 2a). All evaluated floral abnormalities were observed in T-type breeding lines (Figure 1d). Significant differences were found for the presence of *lo* and deformed pistil (*dp*, Figure 2b,c) between D- and T-type cytoplasms (Figure 1d, Appendix A).

Breeding lines with D-type cytoplasm had a significantly higher pollen viability than the breeding lines with T and W/γ cytoplasm types (*p* < 0.001 and < 0.05, respectively) (Figure 1c, Figure 2a–e and Appendix A). This was reflected by the absence of lines with unviable pollen, while over 25% of the T-type breeding lines had unviable pollen. The only genotype for the W/γ-type cytoplasm presented sterile pollen in the distinctive tetrad form of male sterility for the W/γ-type cytoplasm (Figure 2e).

### 2.3. Effects of Cytoplasmic Type on Pollen Tube Germination and Fruit and Seed Production in Test Crosses

The cytoplasmic type significantly affected the proportion of male fertile (1 ≤ L < 2) and sterile (2 ≤ L ≤ 5) lines based on the in vivo pollen tube growth (*p* < 0.001, Appendix A). There were male-fertile and male-sterile lines in both D- and T- cytoplasmic groups, with a higher frequency of male sterile individuals in the cytoplasmic D group.

The cytoplasmic type of the male parent had a significant effect on both fruit and seed production (*p* < 0.001). There was no effect of the female parent on these traits (*p* > 0.1 for both traits, Appendix A). Crosses involving breeding lines with the T-type cytoplasm generally produced significantly more fruits and seeds than those involving breeding lines with D-type cytoplasm (Figure 3b,c). Fruit and seed production were determining factors for concluding whether a breeding line was male fertile or sterile. For these populations, a fruit set percentage (FSP) greater than or equal to 5% and seed set equal to or higher than 20 seeds/berry were considered indicative of male fertility (Appendix A). Based on this cut off, there were 33 male fertile and 38 male sterile lines (46.5 and 53.5%, respectively) in the D-type cytoplasmic group, while the T-type cytoplasmic group had 34 male fertile and 4 male sterile lines (89.5 and 10.5%, respectively).

### 2.4. Correlations among Traits

Pairwise correlations were performed for all the breeding lines and for D- and T-type cytoplasm lines separately. The most significant traits in predicting pollen viability and male sterility or fertility were correlated with fruit set percentage and seed set (Table 2). Anther-lobe overlap (*lo*) had a low but significant correlation with FSP and SS only when evaluating all lines together, but the correlation within each cytoplasmic group was not significant.

Pollen viability (PolV) had a negative weak- however significant- correlation with FSP when evaluating all lines together and a moderate and significant correlation with SS within T-type cytoplasm lines.

In vivo pollen tube growth (IVPTG) was significantly and negatively correlated with FSP and SS when evaluating all lines evaluated together and within T- type cytoplasm. Within D-type cytoplasm only FSP was negatively and significantly was correlated with FSP.

### 2.5. Identification of Potential Fertility Restorer Breeding Lines

Based on the pedigree information of 142 breeding lines and their performance in the test crosses, we could infer 11 parental lines candidate bearers and non-bearers of a fertility restoration mechanism (Table 3, Appendix A).

All progenies of male parental lines CIP392633.64 and CIP396272.43 (both D-type cytoplasm) were male fertile. CIP392633.64 was parent of ten lines from D- and T-type cytoplasms, and CIP396272.43 was parent of six lines, all from T cytoplasmic types (Table 3, Appendix A).

The male parental lines CIP396012.288, CIP392639.2, CIP395017.229 (all D-type cytoplasm) and CIP304372.7 (T-type cytoplasm) generated breeding lines segregating for male fertility/sterility within their same cytoplasmic type (Table 3, Appendix A).

The male parental lines CIP396038.107 (D-type cytoplasm) and CIP396041.102 (T-type cytoplasm) were parents of six and four male sterile breeding lines, respectively, all from D-type cytoplasm (Table 3, Appendix A). Three female parental lines from D-type cytoplasm were involved in all sterile families. These lines can be candidates of *Rf* −.

## 3. Discussion

There is a general tendency of increased frequency of male sterile lines within potato breeding programs due to the increased frequency of T-, D- or W-type cytoplasm [25], and the CIP breeding program is no exception. In a previous study by [14], the B3 late blight resistance breeding population of CIP had the highest proportion of D-type cytoplasm in its breeding lines (81.5%). After seven years of breeding and selection, we found that 96.9% of the genotypes in B3 population carried D-type cytoplasm. The high incidence of D-type cytoplasm relates to CIP’s initial strategy for late blight resistance breeding based on *S. tuberosum* clones with *S. demissum* specific resistance, mainly as female parents [26]. Additionally, D-type cytoplasm is associated with good processing qualities and high starch content, both highly desirable traits selected by breeders [27].

The new late blight resistance population LBHT has a much lower frequency of breeding lines with D-type cytoplasm (38.6%). This is a result of a conscious effort of diversification by combining the high level of late blight resistance in the B3 background with heat tolerant male fertile lines with T-type cytoplasm. Another approach to broaden the base is using landraces from *S. tuberosum* groups Andigenum (*adg*), Phureja or Stenotomum as female parents (the latter two by unreduced eggs or chromosome doubling as in Saikai 35 variety [28]). These species have been commonly used as pollinators in breeding for disease resistance because, when used as females, reductions in yield and late maturity are observed. It has also been proven that fertility restoration within the adg germplasm can be successfully maintained in tuberosum (Tuberosum x Neo tuberosum case) [29,30]. Nevertheless, male fertility is an important factor to consider when selecting parents in a breeding program to avoid crossing failure and low seed set. This is of more significance in a hybrid breeding program where a large quantity of seed per cross is required for sustainable seed production.

Recently Gavrilenko et al. [17] found polymorphic mitochondrial loci related to male fertile phenotype in potato varieties with the W-type cytoplasm. Male fertility gain of function due to mitochondrial polymorphism caused by recombination, rearrangements or mutation has been reported in other Solanaceae crops as tobacco, capsicum and eggplant [31,32,33]. Assuming that in the last 50 years some degree of mtDNA spontaneous mutations may have occurred in CIP breeding lines, D- and T-type cytoplasms were traced back to their 13th maternal lineages identified through pedigree records. Three of the maternal lineages generated the most male sterile families within the D-type breeding lines in this study.

Two inconsistencies were found on derivatives from maternal lineages AC 25953 and INDIA–1058 B. Derivatives of maternal lineage AC 25953 (originally D-type cytoplasm) split into D- and T-types in their pedigree when using clone CIP382171.26 as female parent (approximately four generations backwards), resulting in four breeding lines having the T-type cytoplasm and two breeding lines maintaining the original type (Table 1). Derivatives from INDIA–1058 B (originally T-type) have D-type cytoplasm; the change occurred three generations back when using CIP387415.49, a clone with inaccurate cytoplasmic type records, was used as female parent. Mihovilovich et al. [14] also reported inconsistencies across immediate female parents of their breeding lines in study.

Although floral abnormalities were more frequent in T-type than in D-type breeding lines, they were not strongly correlated with male fertility or sterility. Pollen production and viability resulted in an overall better performance of the D-type cytoplasm group over the T- and W/γ-type groups, indicating an apparent male fertility ranking between cytoplasm types, D > T > W. However, this is in contradiction with the results of the test crosses showing a higher proportion of male fertility in the T-type cytoplasm group, suggesting that pollen production and viability alone should not be considered as indicators of male fertility. Similarly, previous observations reported that D-type cytoplasmic male sterility does not produce abortive pollen grains and look morphologically normal [16,17]. This result may be cultivar-specific since Vanishree et al. [27] reported D-type pollen with several abnormalities at germination such as clumps of inviable pollen, 2–4 pollen germination sites from one pollen grain or curly germination. In our study, pollen viability had a meaningful correlation only with the number of seeds/berry (seed set) in the T-type cytoplasmic group.

It is advisable when evaluating male fertility to use more than one technique [34,35]. In this study we compared pollen viability with in vivo pollen tube growth and fruit and berry set in test crosses. Pollen viability is not a reliable method to predict male fertility in heterogeneous cytoplasmic and nuclear backgrounds. The acetocarmine staining technique´s main advantage is its simplicity, and it can be useful to evaluate individuals within a family or more homogenous groups.

In other Solanaceae hybrid crops as pepper and eggplant, markers for male sterility restoration have been developed. These markers involved the nuclear *ms* genes, *Rf* genes from the *PPR* family and the CMS mitochondrial *atp6* gene [32,36,37]. In potato, *Rf*–*PPR* gene homologues have been identified [21], and markers for the mitochondrial genes *nad2*, *nad1*/*atp6* and *rpl5*–ψ*rps14* have been developed for CMS in W-type cytoplasm.

In this study we propose candidate fertility restorer lines and candidate lines for CMS. These candidates are the parental lines of the current cycle of selection for the three populations evaluated. Even though a segregation of male fertility/sterility was observed in breeding lines already subjected to selection for late blight resistance, yield and tuber qualities, male fertile lines were still (and unintentionally) maintained in the breeding material. The candidate fertility restorer parental lines identified in our study and the progenies developed in the test crosses could be utilized to develop and validate molecular markers that would be extremely valuable for hybrid potato breeding. Such molecular markers will ease the identification of fertility restorer and maintainer lines to create the male and female pools of a reciprocal recurrent selection scheme.

Characterizing the level of cytoplasmic sterility and the plausible presence of a fertility restorer gene in CIP breeding lines is of paramount importance for our diploid hybrid breeding pipeline. Potato diploid hybrid breeding programs reported significant advances on yield, generation of inbred lines, overcoming inbreeding depression and breeding schemes for self-compatibility and agronomic traits fixation [1,6,38,39]. Nevertheless, dihaploids from advanced breeding lines from *S. tuberosum* are mainly male sterile and produce sterile progeny even after *Sli* introgression [7,40,41]. Therefore, specific haploid induction of *Rf* bearers or fertile cytoplasm mutants will enable the hybrid breeding process. Additionally, even though self- and cross-pollinations are still manually conducted, in the future a CMS–*Rf* system will be needed [9].

## 4. Materials and Methods

We evaluated reproductive traits on 142 breeding lines and their male fertility/sterility through test crosses with female lines from different CIP breeding populations. Two separated greenhouses were used to isolate female plants from the male plants and avoid contamination.

### 4.1. Plant Material

Tubers from 142 breeding lines from populations developed for late blight resistance: B3C3, (*n* = 62); LBHT, (*n* = 43); and LBHT x LTVR, (*n* = 37) were grown in screenhouses at CIP Santa Ana experimental station located at decimal degrees (dd) coordinates: −12.01039, −75.22411; 3216 m above sea level. B3C3 population is the third cycle of the recurrent selection scheme of B3 population, which was developed by CIP in the 1990s for late blight resistance. The background of B3 contains *S. tuberosum* modern varieties, *S. tuberosum* Andigenum and Phureja groups and wild species such as *S. demissum*, *S. bulbocastanum* and *S. acaule*. Other important traits in this population are high tuber yield and dry-matter content, early tuberization and bulking, and good quality for potato fries and chips [42,43]. The late blight and heat tolerant (LBHT) population was developed to address climate change in the highlands and the production of the late blight resistant potato. This population results from crossing B3 advanced material with potato virus X (PVX) and Y (PVY) resistant clones and testing them for late blight resistance and heat tolerance [44]. The “lowland tropics virus resistant” (LTVR) population is adapted to warm and arid lowlands, resistant to PVX, PVY and PLRV and mainly composed of T- and W-type cytoplasm [14,45].

The cytoplasmic types determined by Mihovilovich et al. [14] were used as a starting point to trace the cytoplasmic type of the 142 breeding lines used as pollinators. The set of PCR markers used for cytoplasm type determination by Mihovilovich et al. [14] were T, S, SAC, D and A, and the procedure performed was as described by Hosaka and Sanetomo [18].

The above-mentioned breeding lines were used as pollinators for test crosses to evaluate fruit and seed development and yield (see Section 4.4.). Four clones were used as female parents: CIP380389.1 (variety Canchan, T-type cytoplasm, male sterile variety from CIP’s population A); CIP398208.620 (advanced clone of LBHT population, T-type cytoplasm, male fertile line); CIP308520.290 (breeding line, D-type cytoplasm, male fertile line from B3C3 population); and CIP701241 (Huagalina, A-type cytoplasm, male fertile landrace).

### 4.2. Reproductive Trait Evaluation

The experimental unit was one plant (per pot), and each breeding line was sown in two repetitions. Soil preparation, fertilization, transplant and pruning were performed to promote flowering and pollen production [46]. An extended artificial photoperiod of 16 h day/8 h night was applied one month after planting by placing 42 W fluorescent cool daylight bulbs at a 30-cm distance of the canopy level.

The 142 breeding lines were evaluated for basic reproductive traits. Flowering degree (Flwdgr) was evaluated by a visual inspection of three inflorescences per experimental unit and rated in a categorical scale (0–7), where 0 = absence of flower buds; 1 = small flower buds and abscission; 3 = scarce flowering, 2–5 flowers per inflorescence; 5 = moderate flowering, 6–12 flowers per inflorescence; and 7 = profuse flowering with more than 12 flowers per inflorescence. Pollen quantity (QPol) was evaluated by collecting pollen from 10 flowers per experimental unit (plant), and the mean of both repetitions was calculated for each breeding line. Mature anthers were collected and dried at room temperature (~25 °C) for 24 h, then pollen was extracted by vibration and collected in 500 μL Eppendorf tubes. Categorical rating of the collected volume (μL) was done in a scale of 0 to 5, where 0 = absence of pollen, no shedding; 1 = 0–50 μL; 3 = 50–250 μL; and 5 = 250 μL < Pollen viability (PolV) was assessed by acetocarmine staining as described by Ordonez et al. [47] and classified in categories according to their percentage of stainability, i.e., unviable pollen 0–5%; low viability 5–50%; moderate viability 50–80%; and high viability 80–100%.

### 4.3. Floral Abnormalities Evaluation

Floral morphological abnormalities related to cytoplasm type were recorded based on the observations and acronyms proposed by Grun et al. [10] for anther-lobed overlap (*lo*) and anther-style fusion (*ASF*). Additionally, shriveled anthers (Shr) and free and divergent anthers (FD) phenotypes were recorded as proposed by Huaman et al. [48]. Other abnormalities registered in this study were anther-corolla fusion (ACF) and pistil deformities (dp). The presence (1) or absence (0) of each characteristic was scored for each experimental unit during the flowering period, between 60–90 days after transplanting (DAT).

### 4.4. Test Cross Evaluation

The following reproductive traits were evaluated: in vivo pollen tube growth (IVPTG), fruit set percentage (FSP) and seed set (SS). Crosses with breeding lines with unviable pollen were considered as controls.

Pistils were collected 48 h after pollination for in vivo pollen tube growth evaluation. Pistil treatment and staining were performed as described by Ordoñez et al. [47]. A five-level scale (L1–L5) proposed by Camadro and Peloquin [49] was used to rate the length of the pollen tube germination. L1 denotes a compatible cross; the pollen tube reached the ovaries. L2 to L5 are considered as incompatible crosses: In L2, the pollen tube growth was arrested at 3/3 of the length of the style; in L3, the pollen tube growth was arrested at 2/3 of the length of the style; in L4, the pollen tube growth was arrested at 1/3 of the length of the style; and in L5, the pollen tube growth was arrested at stigma level or did not germinate. Two pistils were observed for each cross. The mean values of IVPTG for each breeding line were considered for Wald test and correlations (see Section 4.5). A breeding line was considered as male fertile if 1 ≤ mean IVPTG < 2, and otherwise (mean IVPTG ≥ 2) as male sterile.

A total of 1223 pistil samples from 113 breeding lines used as pollinators were observed. A group of 29 lines was not crossed due to asynchronous flowering with the female counterpart. A total of 1126 pollinations (5554 flowers) was performed for fruit and seed production.

Fruits were harvested 45–60 days after pollination, and seeds were extracted by crushing the fruits in water and filtering the seeds to remove mucilage and debris. The fruit setting percentage (FSP) was calculated as the number of formed fruits divided by the number of pollinated flowers, and the seed set (number of seeds/berry) (SS) was calculated as the total number of seeds divided by the total number of fruits for a given cross [3].

### 4.5. Statistical Analysis and Software

The data were collected using the Fieldbook app [50]. Datasets used in this study are available at CIP Dataverse (Replication Data for: Cytoplasmic Male Sterility Incidence in Potato Breeding Populations with Late Blight Resistance and Identification of Breeding Lines with a Potential Fertility Restorer Mechanism. Available online: https://data.cipotato.org/dataset.xhtml?persistentId=doi:10.21223/80KNLP, accessed on 9 November 2022). FigureJ, a plugin for ImageJ [51], was used for figure assembly.

The non-parametric test of Wilcoxon was used to determine statistically significant differences among cytoplasmic groups D and T for categorical reproductive traits, i.e., flowering degree (Flwdgr) and quantity of pollen (QPol). The only breeding line from W-type cytoplasm was not included in this test because results would not be reliable. For binary traits (floral abnormalities and in vivo pollen tube growth (IVPTG)) where the proportion of a condition are compared, the test for equality of proportions was used to determine differences for cytoplasmic type. All tests were performed in R software 4.2.1 [52].

ASReml–R 4.1.0.130 [53] and asremlPlus 4.3.36 [54] R packages were used to select models and perform pairwise comparisons for the reproductive quantitative traits (pollen viability (PolV), fruit set percentage (FSP) and seed set (SS)). Cytoplasmic type (in all cases) and the female clones (in test crosses) were considered as fixed effects and the breeding lines as random effects. Significance for the fixed effects was assessed using the Wald test.

## 5. Conclusions

There is evidence of CMS and fertility restoration mechanism in the late blight breeding material that leads to the observed low rate of successful crosses. Identified male fertile breeding lines and potential fertility restorers will be further validated and used to identify single nucleotide polymorphisms correlated with the fertility restoration phenotype. Molecular markers for fertility restoration will thus contribute to increasing the rate of successful crosses in our breeding pipelines and, along with candidates for male fertile homozygous line development, will contribute to the implementation of a long-term diploid hybrid breeding with a proper separation of female and male pools, and to discriminating potential male parents before test crosses.

## Figures and Tables

**Figure 1 plants-11-03093-f001:**
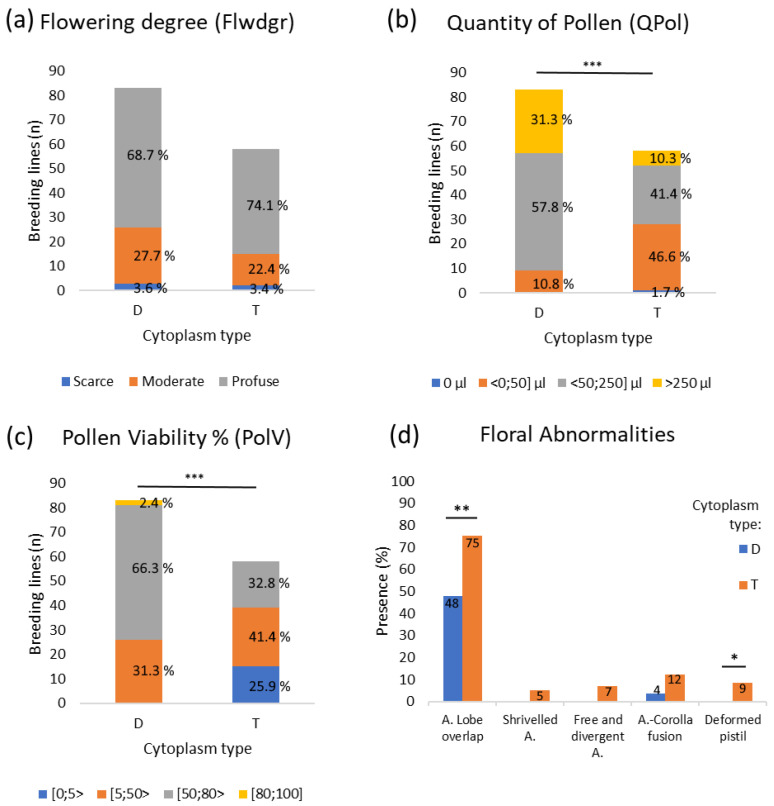
Differences in reproductive traits and floral abnormalities for the D- and T-type cytoplasms. (**a**) Flowering degree (Flwdgr), (**b**) quantity of pollen (QPol), (**c**) pollen viability percentage (PolV), and (**d**) floral abnormalities. Numbers within the bars are the percentage of breeding lines within a cytoplasmic group. Stars show statistically significant differences at *p* < 0.001 (***), *p* < 0.01 (**) and *p* < 0.05 (*) between the cytoplasmic groups for each trait.

**Figure 2 plants-11-03093-f002:**
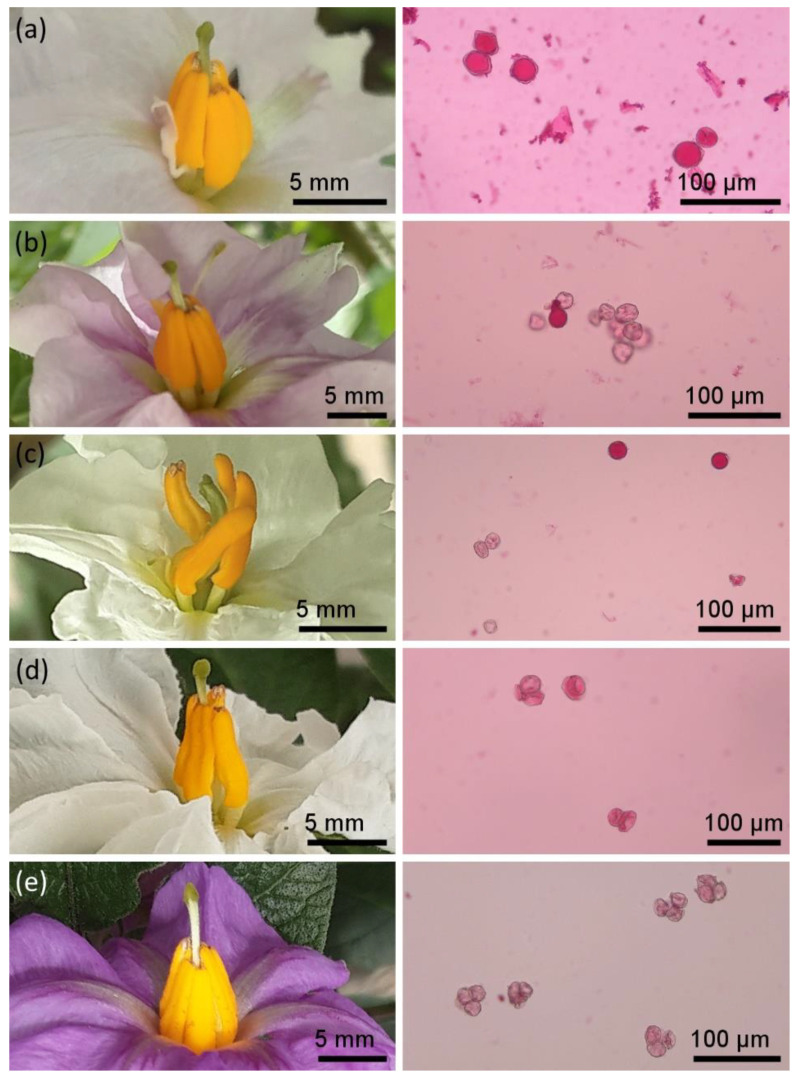
Floral abnormalities and pollen viability (PolV) of advanced late blight resistant breeding lines. D-type cytoplasm line: (**a**) CIP398190.200 showing anther-corolla fusion (ACF), anther-lobe overlap (*lo*) and low PolV. T-type cytoplasm lines: (**b**) CIP398208.33 showing pistil deformities (dp) and low PolV; (**c**) CIP398208.219 showing free and divergent anthers (FD), dp and moderate PolV; and (**d**) CIP398098.231 showing shriveled anthers (Shr) and low PolV. W/γ-type cytoplasm line: (**e**) CIP302506.39 showing *lo* and tetrad sterility.

**Figure 3 plants-11-03093-f003:**
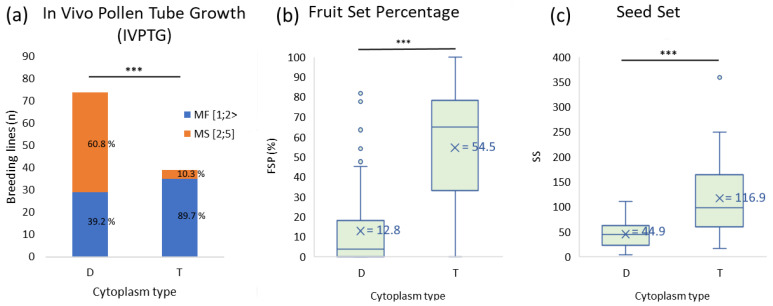
In vivo pollen tube growth (**a**), fruit set percentage (**b**) and seed set (**c**) from test cross evaluations of 113 breeding lines separated by cytoplasm type. In vivo pollen tube growth (IVPTG) shows proportions of male fertile (MF) and male sterile (MS) lines. Stars show statistically significant differences (at *p* < 0.001) between the cytoplasmic groups for each trait.

**Table 1 plants-11-03093-t001:** Cytoplasmic composition and maternal ancestors of three late blight resistance breeding populations.

		Populations		
Oldest Maternal Ancestor	Ancestor Generations ^1^	B3C3	LBHT	LBHT x LTVR
Proportion of D-type breeding lines (N = 83)		96.8%	39.5%	16.2%
AC 25953 ^2^	8	2	--	--
54-Q-2	8	16	--	--
INDIA-832	6	10	3	--
GRETA	6	--	2	2
66-563-13C	6	2	--	--
NEVADA	6	9	11	3
CIP676084	5	13	1	1
INDIA-1058 B ^3^	5	4	--	--
TXY.4	2	4	--	--
Proportion of T-type breeding lines (N = 58)		3.2%	58.1%	83.8%
AC 25953 ^2^	8	--	2	2
CIP800122	7	--	23	29
CIP750627	5	2	--	--
Proportion of W/γ-type breeding lines (N = 1)		0%	2.3%	0%
PG-232	2	--	1	--

^1^ Number of generations back to the oldest maternal records available.^2^ AC 25953 has D-type cytoplasm, nevertheless, is the ancestor of two groups D- and T-type breeding lines. ^3^ INDIA–1058 B has T-type cytoplasm, but all their descendants have D-type cytoplasm.

**Table 2 plants-11-03093-t002:** Overall and cytoplasm-specific correlations between anther-lobe overlap, pollen viability and in vivo pollen tube growth and fruit set percentage and seed set of 109 breeding lines and 5554 crosses.

	D and T Cytoplasm		D Cytoplasm		T Cytoplasm	
	Fruit Set (%)	Seed Set	Fruit Set (%)	Seed Set	Fruit Set (%)	Seed Set
Anther-lobe overlap	0.29 ***	0.23 **	0.10	0.08	0.18	0.04
Pollen viability	−0.17 ***	0.05	−0.10	−0.10	0.18	0.35 ***
In vivo pollen tube growth	−0.53 ***	−0.46 ***	−0.28 ***	−0.16	−0.48 ***	−0.41 ***

Stars show statistical significance at *p* < 0.001 (***) and *p* < 0.01 (**).

**Table 3 plants-11-03093-t003:** Breeding lines with a possible fertility restoration mechanism (*Rf*+) based on the results from the test crosses.

			Breeding Lines						
			D-Type		T-Type		Total MF	Total MS	Candidate
Parent Type	CIP Number	CyT	MF	MS	MF	MS			
Male	CIP392633.64	D	3	0	7	0	10	0	*Rf*+
	CIP396272.43	D	0	0	6	0	6	0	*Rf*+
	CIP304372.7	T	0	1	4	3	4	4	*Rf*+
	CIP392639.2	D	2	2	0	0	2	2	*Rf*+
	CIP395017.229	D	2	2	0	0	2	2	*Rf*+
	CIP396012.288	D	5	3	0	0	5	3	*Rf*+
	CIP396038.107	D	0	6	0	0	0	6	*Rf*−
	CIP396041.102	T	0	4	0	0	0	4	*Rf*−
Female	CIP395117.3	D	--	--	--	--	0	2	*Rf*−
	CIP396034.103	D	--	--	--	--	0	3	*Rf*−
	CIP396046.105	D	--	--	--	--	0	3	*Rf*−

CyT: cytoplasm type, MS: male sterile, MF: male fertile.

## Data Availability

Datasets generated in this study can be found as “Replication Data for: Cytoplasmic Male Sterility Incidence in Potato Breeding Populations with Late Blight Resistance and Identification of Breeding Lines with a Potential Fertility Restorer Mechanism”. Available online: https://data.cipotato.org/dataset.xhtml?persistentId=doi:10.21223/80KNLP (accessed on 9 November 2022).

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
