# Peer review of "Cytoplasmic Male Sterility Incidence in Potato Breeding Populations with Late Blight Resistance and Identification of Breeding Lines with a Potential Fertility Restorer Mechanism"

_plants, 2022, doi:10.3390/plants11223093_

Round 1
Reviewer 1 Report
Potato is the most important vegetative propagated food crops in the world. Traditional potato breeding methods are significantly hampered by a number of genetic factors: autotetraploidy, a high level of heterozygosity, tetrasomic inheritance, inbreeding depression and male sterility of many forms. Developing of the ‘Diploid F1 Hybrid Breeding’ determines the need to study the genetic basis of male sterility in potato including the CMS-Rf genetic system. This study provide characterization of the CMS types and reproductive traits of three CIP’ breeding populations. The main result of this study is identification of male fertile breeding lines and potential Rf-candidate lines. The results obtained have both theoretical and practical significance. I believe that the authors have provided sufficient background and presented the results adequately. At the same time I have some minor issues related to the paper. There are a few comments about the design of the experiments:
-Page 3 - Methods – The experimental unit was one plant (per pot), and each breeding line was sown in two repetitions. Why not in three replicates?
-Page 4 – Methods – Reproductive trait evaluation was evaluated “by collecting pollen from 10 flowers per experimental unit”. It would be more informative to add number of flowers per plant and number of plants per genotype (breeding line) that were used to evaluate different reproductive traits. The authors need to clarify this.
-Page 4 – Methods – “Two pistils were observed for each cross” – is it enough? Is there a standard for Test-cross evaluation? Quote if necessary.
-Page 6 – Results - Table 1 – It is necessary to clarify the “W- type” – what “W-type” cytoplasm does breeding lines have - W/gamma, or W/beta, or W/alpha ??
-Page 12-13 – Discussion - At final of discussion, the conclusions must be described.
Author Response
Dear Reviewer 1: Thank you for your comments, please see the answers in the attachment.

Reviewer 2 Report
Please see attached file.

Author Response
Dear Reviewer 2, thank you for your comments, find the answers in the attachment.
